# Osteoporosis Detection by Physical Function Tests in Resident Health Exams: A Japanese Cohort Survey Randomly Sampled from a Basic Resident Registry

**DOI:** 10.3390/jcm10091896

**Published:** 2021-04-27

**Authors:** Ryuji Osawa, Shota Ikegami, Hiroshi Horiuchi, Ryosuke Tokida, Hiroyuki Kato, Jun Takahashi

**Affiliations:** 1Rehabilitation Center, Shinshu University Hospital, 3-1-1 Asahi, Matsumoto, Nagano 390-8621, Japan; ryuji-osawa@shinshu-u.ac.jp (R.O.); horiuchih@shinshu-u.ac.jp (H.H.); tryosuke@shinshu-u.ac.jp (R.T.); 2Department of Orthopaedic Surgery, Shinshu University School of Medicine, 3-1-1 Asahi, Matsumoto, Nagano 390-8621, Japan; hirokato@shinshu-u.ac.jp (H.K.); jtaka@shinshu-u.ac.jp (J.T.)

**Keywords:** epidemiological study, osteoporosis, osteopenia, locomotive syndrome, two-step test, frailty

## Abstract

Osteoporosis may increase fracture risk and reduce healthy quality of life in older adults. This study aimed to identify an assessment method using physical performance tests to screen for osteoporosis in community dwelling individuals. A total of 168 women aged 50–89 years without diagnosed osteoporosis were randomly selected from the resident registry of a cooperating town for the evaluation of physical characteristics, muscle strength, and several physical performance tests. The most effective combinations of evaluation items to detect osteoporosis (i.e., T-score ≤ −2.5 at the spine or hip) were selected by multivariate analysis and cutoff values were determined by likelihood ratio matrices. Thirty-six women (21.4%) were classified as having osteoporosis. By analyzing combinations of two-step test (TST) score and body mass index (BMI), osteoporosis could be reliably suspected in individuals with TST ≤ 1.30 and BMI ≤ 23.4, TST ≤ 1.32 and BMI ≤ 22.4, TST ≤ 1.34 and BMI ≤ 21.6, or TST < 1.24 and any BMI. Setting cut-off values for TST in combination with BMI represents an easy and possibly effective screening tool for osteoporosis detection in resident health exams.

## 1. Introduction

Osteoporosis is a disorder characterized by a diminution of bone mass and elevated fracture risk [1]. Fractures not only reduce mobility, living function, and quality of life, but also increase mortality and are directly related to life expectancy [2,3]. As the prevalence of older adults in Japan reached 28.1% in 2018 [4], early osteoporosis detection and prevention are important issues in terms of increasing healthy life expectancy and reducing medical costs [5].

Bone mass measurement is considered essential for precisely diagnosing osteoporosis [1]. However, it is difficult to detect bone loss at an early stage due to a lack of subjective symptoms. In their epidemiological study of healthy community members, Lo et al. witnessed that 25.7% of postmenopausal women had untreated osteoporosis [6]. If screening tests can be conducted for community dwelling residents simply and without expensive devices, the number of latent osteoporosis patients may become efficiently reduced by appropriate consultation and management.

Currently, several other simple tools have been established to test for osteoporosis (Female Osteoporosis Self-Assessment Tool for Asia; FOSTA) [7] and the risk of fractures caused by osteoporosis (Fracture Risk Assessment Tool; FRAX) [8].

The present study focused on physical performance as a possible index to easily understand the risk of osteoporosis since previous reports found levels of physical performance and daily activity to be related to bone mineral density (BMD). For instance, Miyakoshi described that women with sarcopenia associated with muscle aging had significantly higher rates of complicating osteoporosis than those who did not [9], and Chan et al. reported that BMD in healthy adolescents had a high correlation with grip strength [10]. Indeed, the relationship between muscle and bone has attracted considerable attention, the mechanism of which is being gradually elucidated [11,12,13].

Several reports have investigated the relationship between physical performance and osteoporosis in the general population [14,15,16,17,18]. However, it remains uncertain which simple physical performance tests better reflect BMD and can be used for osteoporosis screening. Moreover, none have provided reference values for osteoporosis risk for such test items.

For several years, we have been conducting the “Obuse study”, an epidemiology study on locomotor function in older community dwelling people that employs random sampling from the resident registry of a cooperating local government. The purpose of the present investigation is to establish a screening tool for identifying individuals in need of BMD measurement referral during simple physical health examinations.

## 2. Materials and Methods

### 2.1. Participants

The participants in this study were community dwelling women aged 50–89 years who had not been diagnosed as having postmenopausal osteoporosis. As described in a previous report [19], we randomly recruited participants from a local population between October 2014 and June 2017 to build a cohort for an epidemiology study on locomotor function termed the Obuse study. Briefly, we randomly selected men and women from among 11,326 citizens aged 50–89 years who were registered in a cooperating town office and asked them to participate in this study. Requests were made until the number of consenting participants reached approximately 50 for each age group and gender (8 groups: 50 s, 60 s, 70 s, and 80 s of male and female). The final Obuse study cohort contained 415 participants (212 women and 203 men). The participation rate was 32.0%. Of the 212 women, 168 were included in this study after excluding those taking osteoporosis drugs (28 women), receiving hormonal therapy (7 women), premenopausal (6 women), or unable to perform all of the physical performance tests (3 women) (Figure 1).

All participants were surveyed after obtaining written consent based on the Helsinki declaration. This study was conducted after review by our ethics committee (approval number: 2729). The authors declare no conflict of interest of interest.

### 2.2. Osteoporosis Diagnosis

L2-L4 spine (L2-4), bilateral total hip, and bilateral femoral neck BMD were measured using dual-energy X-ray absorptiometry (Prodigy; GE Healthcare, Chicago, IL, USA). The T-scores for each site were calculated based on the manufacturer-provided reference [20]. The smallest T-score was treated as the representative value among the 5 measurement sites. Based on WHO diagnostic criteria, T-score ≥ −1 was classified as healthy, −2.5 < T-score < −1 was classified as osteopenia, and T-score ≤ −2.5 was judged as osteoporosis [1]. FRAX scores were also calculated using patient information and T-scores.

### 2.3. Physical Performance Tests

We measured grip strength, knee extension muscle strength, and one-leg standing time with eyes open. As diagnostic criteria for the recently established locomotive syndrome, two-step test (TST), stand-up test, and Locomo25 scores were evaluated as well [21]. Grip strength in kilograms was determined using a Jamar Hydraulic Hand Dynamometer (Performance Health, Chicago, IL, USA) to obtain the mean values for each side. Knee extension strength was measured with the Leg Extension/Curl Rehab 5530 (HUR, Kokkola, Finland), with measurements taken for both lower limbs, averaged, and divided by body weight (% weight). One-leg standing time was assessed once for each side, with an upper limit of 60 s. The average value of the left and right sides (seconds) was used.

The TST, stand-up test, and Locomo25 are evaluation items for locomotive syndrome proposed by the Japanese Orthopaedic Association. The TST is performed by taking 2 maximum-stride steps and calculating the distance (centimeters) divided by body height (centimeters) [21]. The stand-up test consists of standing up from a sitting position. Participants progressively rise from boxes of 40, 30, 20, and 10 cm in height with both legs or one leg. The tasks were performed in the following order, from easiest to most difficult: both legs 40 cm→30 cm→20 cm→10 cm→one leg 40 cm→30 cm→20 cm→10 cm. The most difficult task completed was used as the subject’s evaluation value. A score of 1 point was allotted for the first task, with 1 additional point given for each subsequent task [21]. Locomo25 is a questionnaire survey consisting of 25 items about pain and difficulties in daily life during the previous month. Each item is graded from 0 to 4 points for a total score of 100 points. A higher score indicates less activity [21]. The questionnaires were mailed to each participant’s home before the screening and collected at the screening venue.

### 2.4. Statistical Analysis

Based on T-score, participants were classified as having healthy BMD, osteopenia, or osteoporosis. Fracture probabilities after 10 years were estimated based on the FRAX computer-based algorithm [8]. The clinical factors of FRAX included age, gender, height, weight, prior fragility fracture, parental history of hip fracture, current smoking habit, glucocorticoid use, rheumatoid arthritis, other causes of secondary osteoporosis, alcohol consumption of 3 units or more per day, and total hip T-score. Next, associations between each physical performance test and T-scores were evaluated by Spearman’s correlation coefficient. Univariate logistic regression analysis was employed to detect physical performance tests related to low BMD. The objective variable was the presence of osteoporosis (i.e., T-score ≤ −2.5), and the explanatory variables were each physical performance test. Next, stepwise logistic regression analysis was performed using the explanatory variable items whose *p*-value was < 0.2 in univariate analysis. This analysis method was chosen to identify factors that were useful for simple screening by clinicians. The best model selected was evaluated by receiver operating characteristic (ROC) curve analysis. If the area under the ROC curve was ≥ 0.7, the combination of exams was considered appropriate for osteoporosis screening. Afterwards, matrices of positive/negative likelihood ratios were constructed for combinations of osteoporosis detection items, whereby a positive likelihood ratio of ≥ 5.0 was considered useful for a suspected diagnosis and a negative likelihood ratio of ≤ 0.2 was judged as useful for an exclusion diagnosis. Likelihood ratios between 0.2 and 5.0 were interpreted as having no screening value. We used R software version 3.6.1 (The R Foundation for Statistical Computing, Vienna, Austria) and EZR Version 2.4-0 [22] for statistical analyses. *p*-values of < 0.05 were considered statistically significant.

## 3. Results

### 3.1. Baseline Data and Osteoporosis Prevalence

Table 1 shows the baseline characteristics of the participants in this study. The mean ± standard deviation age of the cohort was 68.2 ± 10.6 (range: 51–88) years. Height and weight decreased with age, while BMI increased. L2-4 BMD dropped remarkably from the 60 s, and femoral neck BMD was notably low in the 80 s. Table 2 presents the prevalence of osteoporosis by age group. Of the 168 participants, 46 (27.4%) had healthy BMD, 86 (51.2%) had osteopenia, and 36 (21.4%) had osteoporosis. According to FRAX, the incidence of osteoporosis after 10 years by FRAX and the risk of fractures due to falls both increased with age.

### 3.2. Physical Performance Test Results and Correlations with BMD

Physical performance diminished with age, especially in women aged 70 years and above (Table 3). L2-4 T-score had a significant but weak positive correlation with grip strength (Table 4). Femoral neck BMD was significantly correlated with all physical performance tests apart from the stand-up test. Total hip BMD was significantly correlated with all physical performance tests. Both types of femoral BMD exhibited moderate positive correlations with grip strength and TST, while displaying moderate negative associations with age.

### 3.3. Physical Performance Tests Associated with Osteoporosis

Age, BMI, grip strength, one-leg standing, and TST were significantly related factors to osteoporosis in univariate analysis (Table 5). Multivariate analysis revealed significant associations for BMI and TST with osteoporosis (both *p* < 0.01).

### 3.4. Osteoporosis Screening by Physical Performance Tests

Screening for osteoporosis using the combination of BMI and TST was judged as valid by ROC analysis, with an area under the curve of 0.73 (95% confidence interval 0.64–0.82) (Figure 2). Table 6 displays a positive likelihood ratio matrix with incremental values of BMI and TST. For cases of TST ≤ 1.30 and BMI ≤ 23.4, TST ≤ 1.32 and BMI ≤ 22.4, TST ≤ 1.34 and BMI ≤ 21.6, or TST < 1.24 and any BMI, the positive likelihood ratio exceeded 5.0 and osteoporosis could therefore be suspected. On the other hand, no negative likelihood ratios of < 0.2 were detected (Table 7).

## 4. Discussion

According to the results of this study of postmenopausal women aged 50–89 years not treated for bone loss, 21.4% had latent osteoporosis and 51.2% had osteopenia. After adjustment for the age distribution in Japan, the rates of osteopenia and osteoporosis were estimated as 50.5% and 21.9%, respectively. L2-4 BMD correlated significantly with grip strength, while total femur and femoral neck BMD correlated with almost all physical performance tests. The combination of BMI and TST appeared useful to identify possible osteoporosis; this condition may be suspected for TST ≤ 1.30 and BMI ≤ 23.4, TST ≤ 1.32 and BMI ≤ 22.4, TST ≤ 1.34 and BMI ≤ 21.6, or TST < 1.24 regardless of BMI.

We observed that aging affected not only BMD, but also physical performance. Each physical performance test result decreased with age, with marked declines from the age of 70 years. Many studies have described the relationship between muscle and bone [9,10,16,18]. Furthermore, Tachiki et al. [23] reported that maximal muscle strength related to BMD more strongly than did muscle mass because it was an index including stimulation of bone. Similarly in this study, the correlations between many physical performance tests and BMD were considered the result of interactions involving muscle and bone.

Interestingly, TST and Locomo25, which have been used to diagnose locomotive syndrome, showed significant associations with femoral BMD. Locomotive syndrome is a concept defined as a decrease in movement capabilities as proposed by the Japanese Orthopaedic Association [21]. One cause of locomotive syndrome is osteoporosis. No reports have shown a significant association between locomotive syndrome diagnosis and BMD to date. However, lower limb function was found to significantly correlate with locomotion ability, and the amount of activity in daily life was related to BMD [24,25,26]. The present study also supports a relationship between locomotive syndrome and osteoporosis.

Lastly, TST appeared useful as a screening tool for osteoporosis. TST is an examination in which the subject takes two steps at maximum width without losing balance. Earlier studies revealed that TST correlated significantly with such lower limb functions as 6 min walking distance and maximum 10 m walking speed [27] as well as with the ability to perform activities of daily living [28]. Ashe et al. [29] also described that muscle power correlated more strongly with bone density than did maximum muscle tension or muscle mass. Our results suggest that TST may be an indicator of osteoporosis since it can reflect lower limb power more directly than can maximum knee extensor strength or balance of standing on one leg. The influence of TST increased considerably after multivariate analysis, such that the effect of age might have been absorbed. It was also relevant that the cohort’s age range was 50–89 years and did not include young adults with less frequent osteoporosis.

We witnessed that the combination of BMI and TST provided a clinically effective combination of values for osteoporosis screening. Based on the findings in Table 6, there was no single combination of note. However, BMI was a unique value for each participant, with only 1 TST threshold for each participant. TST and BMI can be easily and inexpensively tested anywhere. Using them, it may be possible to encourage residents to undergo osteoporosis testing before symptom onset. On the other hand, no clinically effective negative likelihood ratios were detected in the cohort, indicating that the possibility of osteoporosis in postmenopausal women cannot be excluded by any particular body function test. In the osteoporosis high-risk group with a FOSTA score of less than −4, the positive likelihood ratio was 2.2 and the negative likelihood ratio was 0.5, indicating an inadequacy in detecting osteoporosis.

This study had several limitations. First, there was a deviation in participant selection. Random sampling from a resident registry was presumed as an effective way to construct a study population that faithfully reproduced the target cohort. However, the process of passive participation may have contributed to a high non-participation rate and incomplete removal of extraction bias. Nonetheless, our passive participation method that randomly selected subjects from the general population could create a study group that was more reflective of the actual conditions of community dwelling residents than could an active participation method, by which volunteers were recruited. Another limitation was the existence of regional characteristics. As the local government in our study was located in a suburban area in Japan, which likely differed from the environment of urban areas, our target population might not be qualitatively representative of the general Japanese population and the cut-off range for detecting osteoporosis could be slightly wider. Such regional differences may become more pronounced when race is taken into consideration. The physical characteristics of the cohort could also have limited the study. The small number of patients with very low BMI might have influenced the results; thus, if BMI is higher than 21, it should be assessed in combination with TST for osteoporosis screening. Furthermore, smoking was excluded from the list of factors associated with osteoporosis because few participants smoked. Lastly, this was a small study due to human resource and financial constraints. Larger scale, multiregional surveys are needed.

## 5. Conclusions

In conclusion, in post-menopausal women aged 50–89 years without a diagnosis of osteoporosis, an estimated 21.4% had osteoporosis and 51.2% had osteopenia. More precise BMD measurements are recommended in individuals exhibiting TST ≤ 1.30 and BMI ≤ 23.4, TST ≤ 1.32 and BMI ≤ 22.4, TST ≤ 1.34 and BMI ≤ 21.6, or TST < 1.24 and any BMI in general health exams towards improving healthy life expectancy in community dwelling older adults.

## Figures and Tables

**Figure 1 jcm-10-01896-f001:**
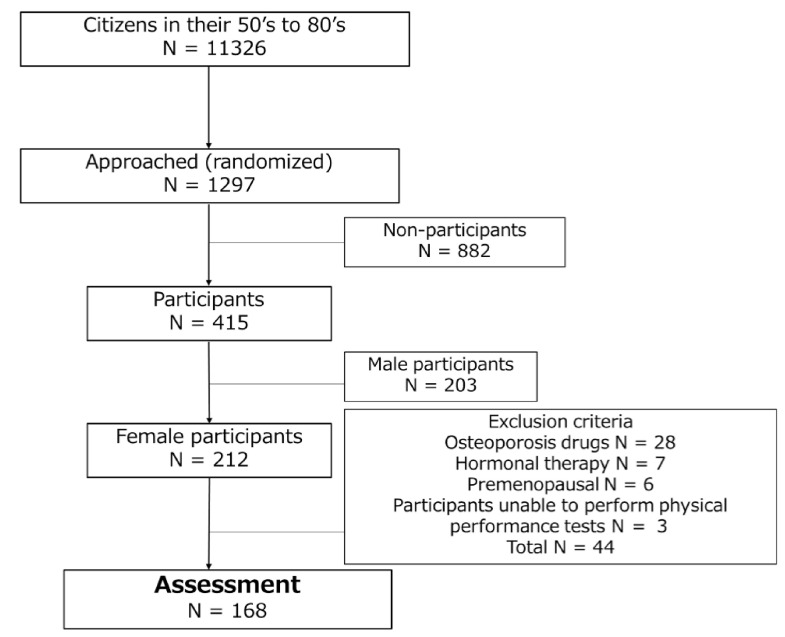
Participant selection flowchart.

**Figure 2 jcm-10-01896-f002:**
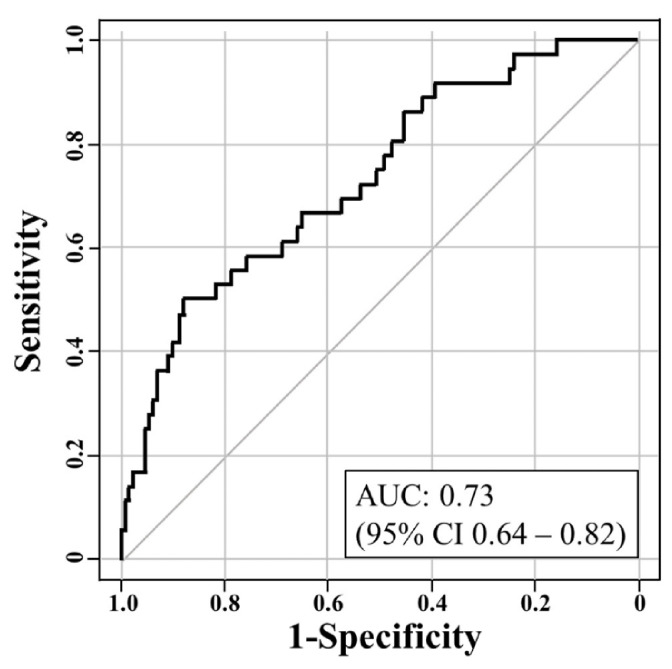
Receiver operating characteristic curve for detecting osteoporosis with the combination of body mass index and two-step test.

**Table 1 jcm-10-01896-t001:** Baseline characteristics of study subjects.

Age Stratum (Years)	N	Height (cm)	Weight (kg)	BMI (kg/m^2^)	L2-4 T-Score	Femoral Neck T-Score	Total Hip T-Score
50–59	42	158.1 ± 5.1	55.0 ± 8.8	22.0 ± 3.9	−0.2 ± 1.5	−0.9 ± 0.9	−0.6 ± 1.1
60–69	58	153.0 ± 5.1	52.4 ± 7.6	22.4 ± 2.8	−0.8 ± 1.5	−1.3 ± 0.9	−1.0 ± 0.9
70–79	39	150.3 ± 5.5	51.6 ± 7.6	22.8 ± 3.1	−0.9 ± 1.6	−1.6 ± 0.9	−1.1 ± 1.0
80–89	29	145.4 ± 5.8	50.4 ± 7.6	23.8 ± 3.1	−0.7 ± 1.8	−2.0 ± 0.7	−1.9 ± 0.8
Total	168	152.3 ± 6.8	52.3 ± 8.0	22.6 ± 3.2	−0.6 ± 1.6	−1.4 ± 0.9	−1.1 ± 1.0

Note: Values are presented as the mean ± standard deviation. Abbreviation: BMI, body mass index.

**Table 2 jcm-10-01896-t002:** Prevalence of osteoporosis and osteopenia and 10 year probability of fractures calculated by FRAX.

Age Stratum (Years)	Normal	Osteopenia	Osteoporosis	FRAX Major Osteoporotic ^†^	FRAX Hip Fracture ^†^
50–59	22 (52.3%)	16 (38.0%)	4 (9.5%)	4.3 ± 1.4	0.5 ± 0.7
60–69	13 (22.4%)	34 (58.6%)	11 (19.0%)	6.9 ± 1.7	0.9 ± 0.7
70–79	9 (23.0%)	19 (48.7%)	11 (28.2%)	9.4 ± 3.7	2.2 ± 2.1
80–89	2 (6.9%)	17 (58.6%)	10 (34.5%)	14.7 ± 4.0	5.8 ± 2.9
Total	46 (27.4%)	86 (51.2%)	36 (21.4%)	8.2 ± 4.4	1.9 ± 2.5

Notes: Values are presented as the number (prevalence). ^†^ The 10 year probability of fracture (%) is expressed as the mean ± standard deviation.

**Table 3 jcm-10-01896-t003:** Results of physical performance tests by age stratum.

Age Stratum (Years)	Grip Strength (kg)	Knee Extension (%Weight)	One-Leg Standing (Sec)	Two-Step Test (No Unit)	Stand-Up Test (Points)	Locomo25 (Points)
50–59	25.1 ± 4.7	1.37 ± 0.36	47.6 ± 15.8	1.54 ± 0.13	4.4 ± 0.9	5.0 ± 4.6
60–69	21.8 ± 3.7	1.22 ± 0.36	44.2 ± 16.7	1.45 ± 0.14	4.4 ± 0.8	4.4 ± 4.6
70–79	20.7 ± 4.2	0.94 ± 0.34	24.6 ± 15.6	1.36 ± 0.19	3.5 ± 1.0	9.9 ± 10.2
80–89	16.9 ± 3.9	0.73 ± 0.36	9.1 ± 9.1	1.05 ± 0.25	3.0 ± 1.0	20.4 ± 15.3
Total	21.5 ± 4.9	1.10 ± 0.42	34.4 ± 20.9	1.38 ± 0.24	4.0 ± 1.1	8.6 ± 10.4

Note: Values are presented as the mean ± standard deviation.

**Table 4 jcm-10-01896-t004:** Correlations between bone mineral density and physical performance.

	L2-4 T-Score		Femoral Neck T-Score		Total Hip T-Score	
	rho	*p*-Value	rho	*p*-Value	rho	*p*-Value
Grip strength	0.24	<0.01 *	0.38	<0.01 *	0.37	<0.01 *
Knee extension	0.03	0.73	0.25	<0.01 *	0.23	<0.01 *
One-leg standing	0.02	0.81	0.31	<0.01 *	0.25	<0.01 *
Two-step test	0.01	0.92	0.39	<0.01 *	0.43	<0.01 *
Stand-up test	−0.11	0.17	0.12	0.10	0.17	0.02 *
Locomo25	0.10	0.18	−0.19	0.01 *	−0.22	<0.01 *
Age	−0.10	0.17	−0.40	<0.01 *	−0.41	<0.01 *
BMI	0.19	0.01 *	0.11	0.17	0.17	0.02 *

Notes: Values represent Spearman’s rho (correlation coefficient). * *p* < 0.05. Abbreviation: BMI, body mass index.

**Table 5 jcm-10-01896-t005:** Physical performance tests related to osteoporosis.

Candidate	Univariate Analysis		Multivariate Analysis	
	Odds Ratio	*p*-Value	Odds Ratio	*p*-Value
Grip strength (−1 kg)	1.09 (1.01–1.19)	0.03 *		
Knee extension (−1%weight)	1.31 (0.53–3.18)	0.55		
One-leg standing (−1 sec)	1.02 (1.01–1.04)	0.01 *		
Two-step test (−1)	10.4 (2.30–46.7)	<0.01 *	31.5 (5.29–188.0)	<0.01 *
Stand-up test (−1 point)	1.12 (0.79–1.59)	0.50		
Locomo25 (+1 point)	1.02 (0.98–1.05)	0.21		
Age (+1 year)	1.06 (1.02–1.10)	<0.01 *		
BMI (−1 kg/m^2^)	1.18 (1.02–1.34)	0.01 *	1.3 (1.11–1.53)	<0.01 *

Notes: Values are presented as the odds ratio (95% confidence interval). * *p* < 0.05. Abbreviation: BMI, body mass index.

**Table 6 jcm-10-01896-t006:** Calculations of positive likelihood ratios for combinations of body mass index and two-step test.

BMI/TST Score	1.24	1.26	1.28	1.3	1.32	1.34	1.36	1.38
**21.0**	22.0	7.3	8.6	7.3	7.3	6.1	5.8	4.5
**21.6**	12.8	6.4	7.3	6.6	6.7	5.8	5.5	4.4
**22.2**	9.8	5.9	5.5	5.2	5.5	4.4	4.0	3.7
**22.4**	11.0	6.6	6.1	5.8	5.3	4.3	3.7	3.4
**23.0**	9.2	6.1	5.8	5.5	4.3	3.4	3.1	2.9
**23.4**	10.1	6.7	6.3	6.0	4.6	3.7	3.1	3.0
**23.8**	5.0	4.0	4.0	4.0	3.4	2.8	2.4	2.4
**24.0**	5.0	4.0	4.0	4.0	3.4	2.8	2.4	2.4

Notes: Leftmost column shows BMI values and top row shows two-step test scores. Values represent positive likelihood ratios. Shaded values indicate ratio ≥ 5.0. Abbreviations: BMI, body mass index; TST, two-step test.

**Table 7 jcm-10-01896-t007:** Calculations of negative likelihood ratios for combinations of body mass index and two-step test.

BMI/TST Score	1.24	1.26	1.28	1.3	1.32	1.34	1.36	1.38
**21.0**	0.8	0.7	0.7	0.7	0.8	0.8	0.7	0.7
**21.6**	0.7	0.7	0.7	0.7	0.7	0.8	0.7	0.6
**22.2**	0.7	0.7	0.7	0.7	0.7	0.7	0.7	0.6
**22.4**	0.7	0.7	0.7	0.7	0.7	0.7	0.6	0.5
**23.0**	0.7	0.7	0.7	0.7	0.7	0.6	0.6	0.5
**23.4**	0.7	0.6	0.6	0.6	0.7	0.6	0.6	0.5
**23.8**	0.7	0.7	0.7	0.7	0.7	0.6	0.6	0.5
**24.0**	0.7	0.7	0.7	0.7	0.7	0.6	0.6	0.5

Notes: Leftmost column shows BMI values and top row shows two-step test scores. Values represent negative likelihood ratios. Abbreviations: BMI, body mass index; TST, two-step test.

## Data Availability

The complete database of the cohort can be accessed at the Zenodo repository (doi.org/10.5281/zenodo.4722536).

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
