# Peer review of "Osteoporosis Detection by Physical Function Tests in Resident Health Exams: A Japanese Cohort Survey Randomly Sampled from a Basic Resident Registry"

_jcm, 2021, doi:10.3390/jcm10091896_

Round 1

Reviewer 1 Report

It's an interesting study. Authors try to detect tools to predict osteoporosis, and help to adequately refer to DEXA. However, they never discuss that some of these tools already exist, such as FRAX algorithm. It should be added.

I have the following concerns and questions, that should be raised : 

  • i do not agree with the extrapolation about percentage of osteoporosis from this study to the whole country population. It should be removed, it's too hazardous.
  • table 1 is not easy to read ...
  • it's surprising that age is not a factor of influence in multivariate analysis. It should be discuss.
  • i understand the ranges, but it seems complicated. Is it possible to determine one alone combination BMI - TST ?
  • it's unclear if BMI alone is not enough to determine osteoporosis
  • osteoporosis is not defined : is it T-score <-2.5 at at least one site ?
  • is it better than FRAX ? is it better than smoking ? is it better than the weight ? Performances of their score should be compared.

Author Response

Question: It's an interesting study. Authors try to detect tools to predict osteoporosis, and help to adequately refer to DEXA. However, they never discuss that some of these tools already exist, such as FRAX algorithm. It should be added.

Answer: Thank you very much for this comment. We have added descriptions of these tools in the Introduction and Discussion.

Revisions:

Paragraph 3 in the Introduction: Currently, several other simple tools have been established to test for osteoporosis (Female Osteoporosis Self-Assessment Tool for Asia; FOSTA) [7] and the risk of fractures caused by osteoporosis (Fracture Risk Assessment Tool; FRAX) [8].

Sentence 6, paragraph 5 in the Discussion: In the osteoporosis high-risk group with a FOSTA score of less than -4, the positive likelihood ratio was 2.2 and the negative likelihood ratio was 0.5, indicating an inadequacy in detecting osteoporosis.

References:

  1. Tong H.; Zong H.; Xu SQ.; Wang XR.; Gong X.; Xu JH.; Cheng M. Osteoporosis Self-Assessment Tool As a Screening Tool for Predicting Osteoporosis in Elderly Chinese Patients With Established Rheumatoid Arthritis. J Clin Densitom. 2019,22(3),321-28.

8. Kanis JA.; Oden A.; Johansson H.; Borgström F.; Ström O.; McCloskey E. FRAX and its applications to clinical practice. Bone, 2009,44(5),734-43.

Question: I have the following concerns and questions, that should be raised :

i do not agree with the extrapolation about percentage of osteoporosis from this study to the whole country population. It should be removed, it's too hazardous.

Answer: We appreciate this advice and have deleted the analysis.

Revision:

We have modified Table 2.

Question: table 1 is not easy to read ...

Answer: We appreciate this comment and have simplified the table.

Revision:

We have modified Table 1.

Question: it's surprising that age is not a factor of influence in multivariate analysis. It should be discuss.

Answer: As the Reviewer states, this result is indeed surprising. We have added a description about it in the Discussion.

Revision:

Sentence 6-7, paragraph 4 in the Discussion: The influence of TST increased considerably after multivariate analysis, such that the effect of age might have been absorbed. It was also relevant that the cohort’s age range was 50-89 years and did not include young adults with less frequent osteoporosis.

Question: i understand the ranges, but it seems complicated. Is it possible to determine one alone combination BMI - TST ?

Answer: Thank you for this comment. According to Table 6, there was no single combination to mention. However, since BMI was a unique value for each participant, there was only one TST threshold for each participant. We have added a description of this.

Revision:

Sentence 2-3, paragraph 5 in the Discussion: Based on the findings in Table 6, there was no single combination of note. However, BMI was a unique value for each participant, with only 1 TST threshold for each participant.

Question: it's unclear if BMI alone is not enough to determine osteoporosis

Answer: We appreciate this question and have added our consideration for low BMI in the Discussion:

Revision:

Sentence 7, paragraph 6 in the Discussion: The small number of patients with very low BMI might have influenced the results; thus, if BMI is higher than 21, it should be assessed in combination with TST for osteoporosis screening.

Question: osteoporosis is not defined : is it T-score <-2.5 at at least one site ?

Answer: Thank you for this question. We have included more detail on the diagnosis of osteoporosis in the Materials and Methods.

Revision:

Sentence 3 in Osteoporosis diagnosis of Materials and Methods: The smallest T-score was treated as the representative value among the 5 measurement sites.

Question: is it better than FRAX ? is it better than smoking ? is it better than the weight ? Performances of their score should be compared.

Answer: Thank you for these important questions. We agree that the outcomes need to be aligned in order to compare scores. In this study, the outcome was BMD because we sought to connect locomotor health examinations with hospital BMD assessment. Since the outcome of FRAX is future fracture risk, the two outcomes cannot be compared directly. However, we have added FRA values to the report. The participants in this study included few smokers, which could not be considered. Body weight was included in the study as BMI.

Revisions:

Sentence 5 in Osteoporosis diagnosis of Materials and Methods: FRAX scores were also calculated using patient information and T-scores.

FRAX results were added to Table 2.

Sentence 6-8, paragraph 6 in the Discussion: The physical characteristics of the cohort could also have limited the study. The small number of patients with very low BMI might have influenced the results; thus, if BMI is higher than 21, it should be assessed in combination with TST for osteoporosis screening. Furthermore, smoking was excluded from the list of factors associated with osteoporosis because few participants smoked.

Sentence 6, paragraph 5 in the Discussion: In the osteoporosis high-risk group with a FOSTA score of less than -4, the positive likelihood ratio was 2.2 and the negative likelihood ratio was 0.5, indicating an inadequacy in detecting osteoporosis.

Reviewer 2 Report

Your definition is solely based on BMD, but a women with a BMD >-2,5 and a low energy vert. / hip fx can also be considered having osteoporosis. How have you sealed with this issue? I guess your study is only useful for diagnosis based on BMD. Please also included this issue into the discussion.

Why have you chosen stepwise logistic regression, since the method has disadvantages?

You wrote: "whereby a positive likelihood ratio of ≥ 5.0 was considered useful for a suspected diagnosis and a negative likelihood ratio of ≤ 0.2 was judged as useful for an exclusion diagnosis." What about the ratio >2 to <5? How have you handled this ratio? What are ratios in this range telling you?

You wrote: "After adjustment for the age distribution in Japan, the rates of osteopenia and osteoporosis were estimated as 50.5% and 21.9%..." Are these numbers reliable? Please discuss epidemiological studies, which reported on osteoporosis, to make it more clear for the reader if you numbers are reliable or over-/underestimated the prevalence. 

Please add as limitation, that your study sample was very small.

Author Response

Question: Your definition is solely based on BMD, but a women with a BMD >-2,5 and a low energy vert. / hip fx can also be considered having osteoporosis. How have you sealed with this issue? I guess your study is only useful for diagnosis based on BMD. Please also included this issue into the discussion.

Answer: We appreciate this comment. As the Reviewer pointed out, the definition of osteoporosis may be insufficient with BMD alone in the clinical context. However, the only common international diagnostic criterion for osteoporosis is the WHO BMD cutoff. In Japan, major osteoporotic fractures are included in the definition of osteoporosis without BMD assessment. We included only undiagnosed subjects in this study, and aimed to identify individuals in need of BMD measurement. This has been added to the Introduction.

Revision:

Sentence 2, paragraph 6 in the Introduction: The purpose of the present investigation is to establish a screening tool for identifying individuals in need of BMD measurement referral during simple physical health examinations.

Question: Why have you chosen stepwise logistic regression, since the method has disadvantages?

Answer: Thank you for this question. Indeed, stepwise logistic regression analysis has the disadvantage that clinically important factors may not always be included in the final model since it is based on mathematical choices. In the present study, this analysis method was chosen in order to discover factors that were useful for simple screening by clinicians. We have clarified our use of the test in the Materials and Methods.

Revision:

Sentence 7 in Statistical analysis of Materials and Methods: This analysis method was chosen to identify factors that were useful for simple screening by clinicians.

Question: You wrote: "whereby a positive likelihood ratio of ≥ 5.0 was considered useful for a suspected diagnosis and a negative likelihood ratio of ≤ 0.2 was judged as useful for an exclusion diagnosis." What about the ratio >2 to <5? How have you handled this ratio? What are ratios in this range telling you?

Answer: Thank you for raising this point. We have clarified this range in the Materials and Methods.

Revision:

Sentence 11 in Statistical analysis of Materials and Methods: Likelihood ratios between 0.2 and 5.0 were interpreted as having no screening value.

Question: You wrote: "After adjustment for the age distribution in Japan, the rates of osteopenia and osteoporosis were estimated as 50.5% and 21.9%..." Are these numbers reliable? Please discuss epidemiological studies, which reported on osteoporosis, to make it more clear for the reader if you numbers are reliable or over-/underestimated the prevalence.

Answer: We appreciate this comment and have removed the analysis at the recommendation of the other Reviewer.

Question: Please add as limitation, that your study sample was very small.

Answer: As advised, we have added this point as a study limitation.

Revision:

Sentence 10-11, Paragraph 6 in the Discussion: Lastly, this was a small study due to human resource and financial constraints. Larger-scale, multi-regional surveys are needed.

Round 2

Reviewer 1 Report

All the issues addressed has been answered

Reviewer 2 Report

Thank you for revising the manuscript.